# Bacterial Cellulose/Tomato Puree Edible Films as Moisture Barrier Structures in Multicomponent Foods

**DOI:** 10.3390/foods11152336

**Published:** 2022-08-05

**Authors:** John A. M. Freitas, Giovana M. N. Mendonça, Leticia B. Santos, Jovan D. Alonso, Juliana F. Mendes, Hernane S. Barud, Henriette M. C. Azeredo

**Affiliations:** 1Graduate Program in Biotechnology, Federal University of São Carlos (UFSCar), São Carlos 13565-905, Brazil; johnallef5@gmail.com; 2Graduate Program in Food, Nutrition and Food Engineering, UNESP—São Paulo State University, Araraquara 14800-903, Brazil; giovana_mendonca@hotmail.com (G.M.N.M.); leticia.bueno@unesp.br (L.B.S.); 3Institute of Chemistry of Araraquara, UNESP—São Paulo State University, São Paulo 01049-010, Brazil; alonso.jdr@gmail.com; 4Embrapa Instrumentation, São Carlos 13560-970, Brazil; julianafarinassi@gmail.com; 5Biopolymers and Biomaterials Group, University of Araraquara (UNIARA), Araraquara 14801-340, Brazil; hernane.barud@gmail.com

**Keywords:** edible films, multicomponent foods, sensory properties, texture, biopolymers

## Abstract

Edible films have been studied mainly as primary packaging materials, but they may be used as barrier layers between food components, e.g., by reducing the moisture migration between components with contrasting water activities. Since edible films are part of the food itself, components adding sensory appeal (e.g., fruit purees) are usually desirable. The objective of this study was to develop a film to be applied as a moisture barrier between nachos and guacamole. Ten film formulations were prepared according to a simplex centroid design with three components—a polysaccharide matrix (consisting of a 5:1 mixture of bacterial cellulose—BC—and carboxymethyl cellulose), tomato puree (for sensory appeal), and palm olein (to reduce hydrophilicity)—and produced by bench casting. The film with the highest palm olein content (20%) presented the lowest water vapor permeability, and its formulation was used to produce a film by continuous casting. The film was applied as a layer between nachos and guacamole, and presented to 80 panelists. The film-containing snack was preferred and considered as crispier when compared to the snack without the film, suggesting that the film was effective in reducing the moisture migration from the moist guacamole to the crispy nachos.

## 1. Introduction

Edible films are thin layers formed from food-grade components, including a macromolecule (usually a polysaccharide or protein) forming the film matrix, and usually containing other components to improve the tensile properties of the final materials, including plasticizers (which are small molecules that are dispersed between the chains of the matrix, increasing free volume, and enhancing film elongation). Fruit and/or vegetable purees may be added to edible film formulations, acting not only to add sensory appeal to the edible materials, but also to incorporate nutritional/functional properties, and also contributing to the film formation, since those purees usually have film-forming polysaccharides (e.g., pectin, starch) as well as plasticizing sugars and organic acids [1]. Different fruit purees have been already included in films, such as papaya [2,3], cupuassu [4], Indian jujube [5], guava, and mango [6]. 

Although edible films have been mainly studied as primary food packaging materials that may be consumed with the food, they have other applications not directly related to food packaging, including moisture barrier layers to be applied between components with contrasting water activities (A_w_) in multicomponent foods such as pizzas, wraps, and tacos [7]. In those foods, the parts with low A_w_ tend to become soggy when in contact with the parts with high A_w_, with the consequent loss of their typical crispy or firm texture. Thus, the shelf life of those products depends on the speed of the water transfer between the components [8]. This may be an issue not only for industrialized food products, but also for food items to be prepared and home-delivered by food service establishments when the time between preparation and consumption may be enough for moisture migration and loss of texture. O’Connor et al. [7] have demonstrated the positive effects of different edible films on reducing the softening of bread crumbs induced by moisture migration from cheese slices.

Bacterial cellulose (BC) is considered as one of three main classes of nanocelluloses, along with cellulose nanocrystals and cellulose nanofibrils [9]. Besides a variety of uses (mainly in the biomedical field) [9,10], BC has been used as a polysaccharide matrix in edible films [6,11,12]. Since BC is produced by bacteria as a membrane, it needs to be disintegrated in order to be used in a powder (or suspension) form for further formulations, as for most food applications [13]. However, the redispersion of powdered BC in water is impaired by hornification (the irreversible aggregation of cellulose chains upon drying) [14]. The problem is usually dealt with either by chemically modifying BC by introducing electrically charged groups (e.g., carboxyl groups inserted by oxidation) [14], or by combining BC to an electrically charged polymer such as carboxymethyl cellulose (CMC), thus promoting electrostatic repulsion and water dispersibility [15,16].

However, as with most polysaccharide-based films, its structure is mainly hydrophilic, causing its films to have relatively high-water vapor permeability (WVP). A common approach to reduce the WVP of polysaccharide-based films is to add a hydrophobic phase (usually a lipid) emulsified into the film structure, either as a conventional surfactant-stabilized emulsion [17,18] or as a Pickering emulsion stabilized by colloid particles [19,20]. 

The objective of this study was to produce a film to reduce the moisture migration in a multicomponent food model consisting of nachos covered with a guacamole spread. Films were formulated with different proportions between BC (as film matrix and Pickering emulsion stabilizer), tomato puree (as the sensory-appealing component), and palm olein (as a hydrophobic agent). The film characterization included Fourier-transform infrared (FTIR) spectroscopy, scanning electron microscopy, water vapor permeability, water contact angles, and shear force measurements.

## 2. Materials and Methods

### 2.1. Film Preparation (by Bench Casting)

BC membranes produced by the bacteria *Komagataeibacter rhaeticus*, previously purified and dried, were provided by Biosmart Nanotechnology (Araraquara, São Paulo, Brazil). The use of the bacteria was registered in the Brazilian National System of Management of Genetic Heritage and Associated Traditional Knowledge (SisGen) under registration number A8C1372. The membranes were milled in an analytical mill (A11, Ika Werke GmbH, Staufen, Germany) for 10 min then ground into a powder in a ball mill (CT-241, Servitech, Santa Gertrudes, São Paulo, Brazil) for 20 min at 380 rpm. The BC powder was then combined to CMC (Sigma 419303) at a BC:CMC weight ratio of 5:1 before resuspension in water.

The films were formulated according to a simplex centroid design (Figure 1) with three components, namely: polysaccharides (BC + CMC), tomato puree (Fugini Alimentos, Monte Alto, São Paulo, Brazil), and palm olein (180 A, Agropalma, Limeira, São Paulo, Brazil). 

All formulations were prepared in such an amount as to contain 25 g of solids (combining polysaccharides and tomato puree solids (the tomato puree used in this study presented a solid content of 17 wt%)) for a total volume of 250 mL; that is to say, the water volumes varied for each formulation in order to meet this requirement. For each formulation, the BC/CMC mixture was firstly suspended (at 2 wt%) in distilled water, in an Ultra-Turrax T18 (Ika Werke, Staufen, Germany) at 16,000 rpm for 30 min. The palm olein was then added dropwise, keeping the Ultra-Turrax homogenization for another 10 min. The tomato puree was then added along with glycerol (plasticizer, at 45 wt% on the dry weight of BC + CMC), keeping the homogenization for an extra 10 min. The dispersions were then vacuum degassed at 800 mbar for 90 min, cast onto glass plates covered with a polyester (Mylar^®^) film, leveled with a draw-down bar for a wet layer thickness of 2 mm, and dried at 50 °C for 24 h. 

### 2.2. Film Characterization

The dried films were detached from the Mylar substrate, cut into test samples (in dimensions as required for each determination), and conditioned for 48 h at 25 °C and 50% RH in an Ethik 420-2TS climatic chamber (Nova Ética, Vargem Grande Paulista, Brazil) prior to the analyses. The thicknesses of the test samples (10 measurements for specimen) were measured by using a KR1250 coating thickness tester (Akrom, São Leopoldo, Brazil) to the nearest 1 μm.

#### 2.2.1. Fourier-Transform Infrared (FTIR) Spectroscopy

The FTIR spectra were recorded with a Vertex 70 FTIR spectrometer (Bruker, Ettlingen, Germany) in attenuated total reflectance (ATR) mode equipped with a diamond crystal accessory. The spectra were scanned over the 4000–500 cm^−1^ wavelength range, with a 4 cm^−1^ resolution over 32 scans. 

#### 2.2.2. Scanning Electron Microscopy (SEM)

The SEM was conducted by using a JSM 6510 microscope (Jeol, Tokyo, Japan). Film samples were mounted on aluminum stubs with their film-air surfaces (as upon drying) facing upward, using conductive carbon tape, then sputter-coated with a 10 nm-thick gold layer with an SCD 050 Sputter Coater (Leica Microsystems, Wetzlar, Germany). Other specimens were immersed in liquid N_2_ for 5 min, fractured with tweezers, and mounted on aluminum stubs with the fractured surface facing upward, using conductive carbon tape, then sputter-coated with a 10 nm-thick gold layer. The samples were examined using an accelerating voltage of 5 kV at 250× magnification for surfaces, and 10 kV at 370× for cross-sections.

#### 2.2.3. Water Vapor Permeability (WVP)

The WVP determination, with ten replicates, was based on the ASTM method E96/E96M-16 (ASTM, 2016) at 25 °C, by using polytetrafluoroethylene (PTFE) permeation cells (24 mm in diameter and 10 mm in height) containing 1.5 mL of distilled water (100% RH) in an Ethik 420-2TS climatic chamber at 25 °C and 50% RH. The cells were weighed at least eight times (in intervals higher than 1 h) within about 24 h. The WVP values were calculated according to the following equation:
(1)
$$WVP=\frac{∆w\times L}{∆t\times A\times \left(p_{1}-p_{2}\right)},$$

where Δ*w*/Δ*t* is the weight loss per time; *L* is the film thickness; *A* is the exposed area of the film; and p_1_ and p_2_ are the partial vapor pressures of water vapor on the inside and outside of the test film, respectively, given as *p*_1_ = P_sat_ × RH_in_ and *p*_2_ = P_sat_ × RH_out_. RH_in_ and RH_out_ are the relative humidities of the internal (~100%) and external sides (test RH) of the film, and P_sat_ is the saturated vapor pressure at the test temperature. 

#### 2.2.4. Water Solubility

The water solubility was carried out in quadruplicate, based on the method proposed elsewhere [21]. Previously dried (105 °C, 24 h) and weighed film disks (4 cm in diameter) were immersed into 50 mL distilled water for 24 h under stirring (100 rpm) at 25 °C in a shaker. The dry weight of the remaining film pieces (if any), obtained after filtration on previously dried and weighed filter paper, was used to calculate the water solubility as a percentage of the initial dry weight of the film (g/100 g).

#### 2.2.5. Shear Force

The shear force test, carried out to simulate the action of incisor teeth [22], was performed (with ten replicates) by using a Knife Edge blade and slotted base (HDP/BS, Stable Micro Systems Ltd., Godalming, UK), with a speed of 2 mm s^−1^, in order to determine the force required to cross-cut each fragment. The films were cut into 5 × 2 cm samples. The results were expressed as the peak force, in Newtons (N).

#### 2.2.6. Water Contact Angles

Water contact angles were determined (with seven replicates) using a CAM 101 Optical Contact Angle Meter (KSV Instruments, Helsinki, Finland) equipped with a CCD KSV-5000 camera and KSV CAM 2009 software (KSV Instruments, Helsinki, Finland). A 4-µL drop of Milli-Q water was placed on the film surface, and images were registered at 0.4 s.

### 2.3. Statistical Analyses

The results were analyzed using the software Minitab^®^ (Minitab Inc., State College, PA, USA). The model adjustment was made in terms of pseudo-components. The best treatment was chosen according to the film properties, mainly on the minimization of WVP, since the film was meant to reduce moisture migration in multicomponent foods. 

### 2.4. Continuous Casting

A film was then produced (from the chosen film formulation) in a continuous casting device (KTF-B, Werner Mathis AG, Zürich, Switzerland). The film-forming dispersion was prepared as previously described for bench casting, except that, in order to avoid the dispersion from running off the substrate, the film-forming viscosity was adjusted by doubling its solid content (500 g solids for 2500 mL of dispersion). The dispersion was continuously (manually) spread onto a Mylar substrate moving through a conveyor, with the wet layer thickness being adjusted to 0.8 mm with a B-type doctor blade. The wet layer then passed through two consecutive 80 cm-long drying chambers with a controlled temperature (80 °C), at a speed of 10 cm min^−1^. 

### 2.5. Sensory Analysis

The film made by continuous casting was sensorially evaluated as a moisture barrier in snacks consisting of Doritos^®^ corn nachos (Pepsico Brasil, Itu, Brazil) and home-prepared guacamole. In the test treatment, each nacho was covered with a film sample (for its whole area) and 4 g of guacamole. A control treatment was made as well, in which the nachos received only the guacamole (without the film). All snacks were prepared 30 min before being evaluated by the panelists, in order to simulate the time taken between snack preparation and consumption (including delivery time). The analysis was carried out in individual booths with red light (to mask the presence of the films), with 80 panelists with ages ranging from 18 to 70 years. Each panelist received a tray with both samples (test and control) in randomized order, with each sample codified with random 3-digit numbers. The panelists were required to fill out a form with three parts. In the first part, they were asked about how much they liked each sample (according to a 9-point structured hedonic scale ranging from 1—“extremely disliked”—to 9—“extremely liked”). The second part was a paired preference test, in which they were asked which sample they preferred. Finally, the third part was a paired crispiness comparison, in which they were asked which sample was crispier. The averages of the acceptance test (hedonic scale) were compared by a paired *t*-test, while the paired comparisons (on preference and crispiness) were evaluated by comparing the number of responses for the most frequently chosen sample with the minimum value required to establish a significant difference, according to the number of panelists [23]. The study was approved by the Research Ethics Committee of the Federal University of São Carlos (UFSCar) (CAAE n. 18034919.0.0000.5504). 

## 3. Results and Discussion

### 3.1. FTIR

The FTIR spectra were presented for films #1, #2, and #3 (which are the vertices of the triangle on the experimental design), as well as the main film components (Figure 2). There were no noticeable differences in the spectra of different films. The broad band around 3300 cm^–1^ is ascribed to the stretching of the OH groups [24]. The C–O–C stretching at 1161 cm^−1^ is ascribed to β(1,4)-glycosidic linkages in cellulose [25] and the presence of esters in palm olein [26]. There are other bands related to cellulose, including the C–O–C asymmetric stretching of the cellulose rings [27], C–H rocking at the β-glycosidic linkage at 893 cm^−1^ [28], and the O–H out-of-phase bending at 667 cm^−1^ [29]. Some bands are related to vegetable oils (including palm olein), such as the ones at 2920 and 2853 cm^−1^ corresponding to the symmetric and asymmetric stretching of aliphatic CH_2_ groups, the C=O stretching band of ester carbonyl groups of triglycerides at 1743 cm^−1^, and the bending vibration of CH_2_ and CH_3_ aliphatic groups at 1462 cm^−1^ [30]. The bands at 1010–1030 cm^−1^ (present in the spectra of the polysaccharides and tomato puree) are ascribed to C-C and C-O stretching [31]. 

### 3.2. SEM

The SEM micrographs of the films (Figure 3) presented rough surfaces and cross-sections, due to the heterogeneity of the film compositions. The film #3 (containing the highest palm olein content) presented noticeable discontinuities (especially on the cross-section), ascribed to the oil droplets within the film structure, which may explain the reduced shear forces when the palm olein contents increased (Table 1, Figure 4). 

### 3.3. Water Vapor Permeability (WVP), Water Contact Angle (WCA), and Shear Force

The water solubility values of the films were not presented, since the films were completely disintegrated after 24 h of immersion in water, with no remaining film pieces. 

The WVP, WCA, and shear force values of the films resulting from different formulations, as well as the corresponding estimated regression coefficients, are presented at Table 1, while Figure 4 shows the graphical representation of the statistical models. 

Palm olein was the component that most contributed to lowering the WVP and to increasing the contact angles (Table 1, Figure 4), which is ascribed to the hydrophobicity of palm olein. Indeed, other studies reported the effects of vegetable oils on reducing WVP and increasing the contact angles of films [32,33,34]. Moreover, palm olein contributed to decrease the shear forces (Table 1, Figure 4). If this study was focused on improving the mechanical strength of the material itself, higher shear forces would be desirable (for probably correlating with better tensile properties). For the purposes of this study, on the other hand, films with lower shear forces were regarded as more interesting, since they are expected to be less perceptible (in terms of texture) when inserted between food components. So, the film formulation #3 was chosen as the best alternative in terms of acting as a moisture barrier in multicomponent foods, due to its relatively low WVP and high contact angle, while also being expected to be less noticeable in terms of texture. 

### 3.4. Film Appeareance and Sensory Analysis

The film produced by continuous casting (from film formulation #3) exhibited an orange color (Figure 5) and was cut into triangles in order to be applied over the nachos on sensory evaluation. The results from the sensory evaluation (Table 2) indicated that the nacho-based snacks containing the film between the nachos and the guacamole layer were significantly more accepted (in a hedonic scale) than the snacks without the films. The film-containing snacks were also significantly preferred (according to the paired preference test), and considered as crispier, indicating that the film was effective in reducing the moisture migration from the guacamole to the nacho and also to improve the snack acceptability. 

## 4. Conclusions

The film containing a combination of tomato puree, polysaccharides (namely, bacterial cellulose and carboxymethyl cellulose, 5:1 weight ratio), and palm olein at a weight ratio of 5:3:2 presented a relatively low water vapor permeability and was then tested as a moisture barrier layer between nachos and guacamole. The results from the sensory test suggested that the film reduced the moisture migration from the guacamole to the nachos, thus helping to maintain the typical nacho crispiness after 30 min. The film prepared in this study may also be used as a moisture barrier layer in other multicomponent foods, such as in pizzas (between the crust and the toppings). 

## Figures and Tables

**Figure 1 foods-11-02336-f001:**
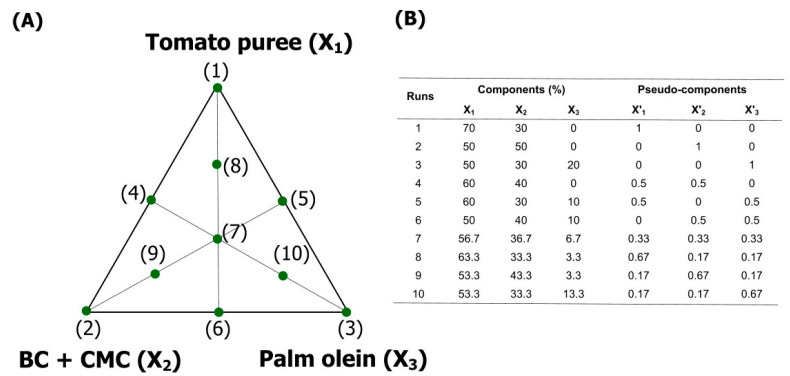
Simplex centroid design. (**A**) Graphical representation of the experimental runs (film formulations). (**B**) Conditions (as components and pseudo-components) of each run.

**Figure 2 foods-11-02336-f002:**
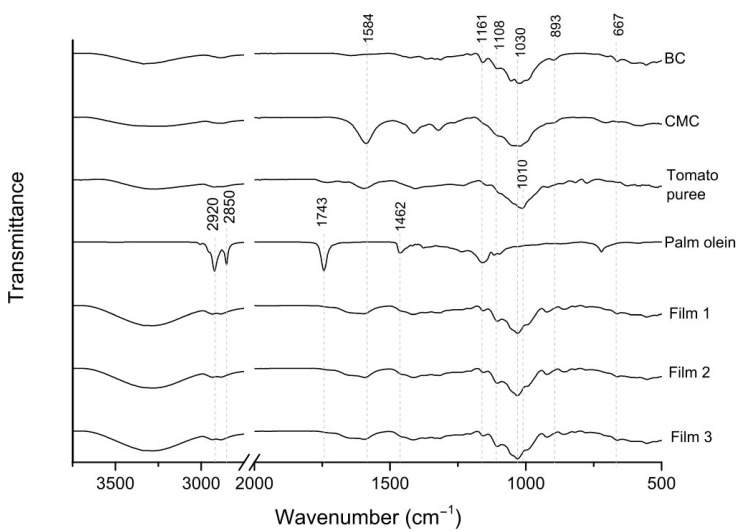
FTIR spectra of film components and films.

**Figure 3 foods-11-02336-f003:**
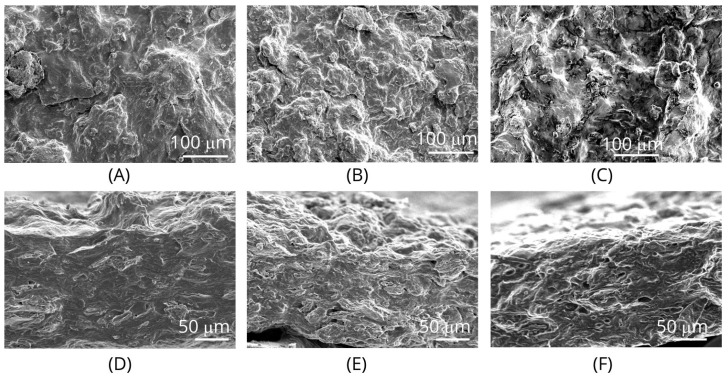
SEM micrographs of films: (**A**–**C**): surfaces of films #1, #2, and #3 respectively; (**D**–**F**): cross-sections of films #1, #2, and #3 respectively.

**Figure 4 foods-11-02336-f004:**
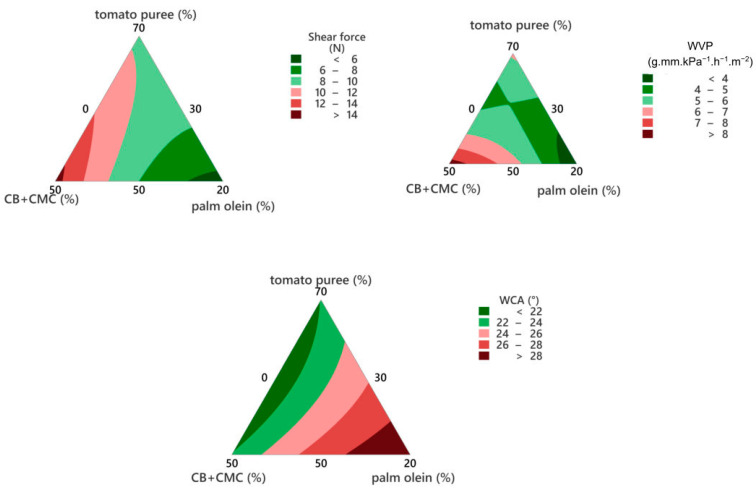
Contour plots of the film properties. WVP: water vapor permeability; WCA: water contact angle.

**Figure 5 foods-11-02336-f005:**
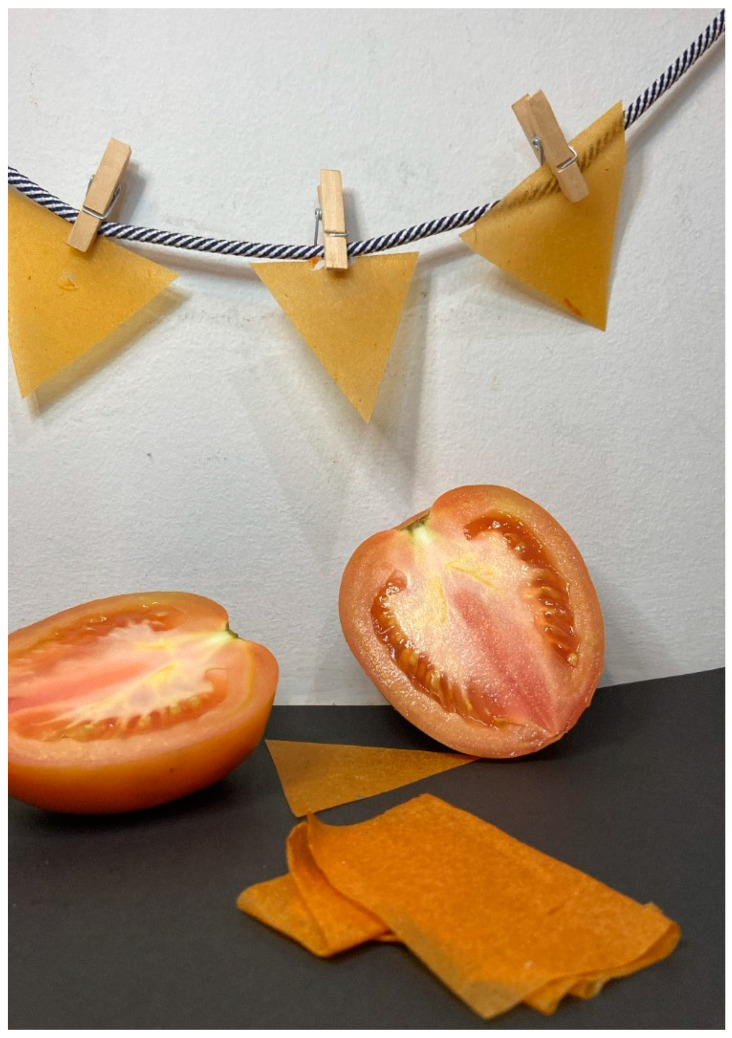
Visual appearance of the film produced by continuous casting.

**Table 1 foods-11-02336-t001:** Properties of films from different formulations and regression coefficients (in pseudo-components).

Formulation	WVP (g.mm.kPa^−1^ h^−1^ m^−2^)	WCA (°)	Shear force (N)
1	5.97 ^c^	22.7 ^b^	10.3 ^b,c^
2	8.28 ^a^	22.0 ^b^	14.7 ^a^
3	3.24 ^e^	29.9 ^a^	5.04 ^e^
4	4.55 ^d^	20.9 ^b^	12.2 ^a,b^
5	4.67 ^d^	26.4 ^a,b^	8.65 ^c,d^
6	6.44 ^b,c^	26.9 ^a,b^	7.43 ^d,e^
7	4.74 ^d^	25.7 ^a,b^	10.1 ^b,c^
8	5.98 ^b,c^	20.5 ^b^	7.96 ^d^
9	7.08 ^b^	23.2 ^a,b^	11.4 ^b^
10	3.92 ^d,e^	26.8 ^a,b^	8.31 ^c,d^
**Terms**	**Coefficients**		
	**WVP**	**WCA**	**Shear force**
β_1_	6.21	22.08	9.67
β_2_	8.44	22.13	14.7
β_3_	2.99	29.83	5.41
β_12_	−9.39	−6.30	−1.47
β_13_	0.32	−0.85	4.36
β_23_	2.66	4.20	−8.33
R^2^ (%)	90.38	90.39	90.24
F	7.52	7.53	7.40
P	0.04	0.04	0.04

Values (for the same property) followed by at least one common letter are not significantly different from each other (Tukey, *p* > 0.05). WVP: water vapor permeability; WCA: water contact angle. β_1_, β_2_, β_3_, β_12_, β_13_, β_23_: terms for tomato solids, polysaccharides (BC + CMC, at a 5:1 weight ratio), palm olein, and interactions (tomato × polysaccharides, tomato*palm olein, and polysaccharides*palm olein), respectively (as pseudo-components).

**Table 2 foods-11-02336-t002:** Results from sensory evaluation of nachos covered with guacamole (NG) or with film (film #3, produced by continuous casting) and guacamole (NFG).

Samples	Acceptance *	Number of Favorable Responses on Paired Preference Test	Number of Responses as “Crispier” ^‡^
NG	7.70 ± 1.36	23	26
NFG	8.03 ± 1.27	**57**	**52**

* Acceptance values on a 9-point structured scale (from 1—“extremely disliked” to 9—“extremely liked”); values presented as means ± standard deviations. ^‡^ Two panelists answered that they did not notice differences in crispiness between the samples. Bold values for the paired preference and crispiness comparison represent significant preference/difference (values higher than 50, presented as the lowest required value for a significant preference in a sensory panel with 80 panelists, according to Lawless and Heymann) [23].

## Data Availability

The data presented in this study are available on request from the corresponding author. The data are not publicly available due to them containing information that could compromise research participant privacy.

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
