# Peer review of "Bacterial Cellulose/Tomato Puree Edible Films as Moisture Barrier Structures in Multicomponent Foods"

_foods, 2022, doi:10.3390/foods11152336_

Round 1

Reviewer 1 Report

The manuscript intitled "Bacterial cellulose/tomato puree edible films as moisture barrier structures in multicomponent foods" is an original research study particularly good organized but not so good presented. The authors must improve their quality of presentation of such interesting results to achieve the high levels of Foods/MDPI journal.

1. In Introduction the methods used for the characterization and testing of obtained films must be included in the last paragraph.

2. Materials and Methods: The authors must separate in paragraphs this section. For example: 2.3 FTIR, 2.4 SEM, 2.5 WVP etc.

3. For WVP which is the equation used for the calculated values. PLs refer into the revised materials and results section.

4. Make a seperate materials and method section for the statistical analysis used.

5. Consider the addition of some antioxidant activity test of films (DPPH assay) as the tomato puree used gives some antioxidant activity to such films.

6. Results and discussion: This section must also separate in section and re-organized to be more readable for readers. So, start first with FTIR and SEM characterization result section and continue with the other experimental results.

Best wishes!

Author Response

Response – Reviewer 1:

The manuscript intitled "Bacterial cellulose/tomato puree edible films as moisture barrier structures in multicomponent foods" is an original research study particularly good organized but not so good presented. The authors must improve their quality of presentation of such interesting results to achieve the high levels of Foods/MDPI journal.

The authors thank the reviewer for the positive comments. We hope that the presentation is satisfactory in this revised version.

  1. In Introduction the methods used for the characterization and testing of obtained films must be included in the last paragraph.

The following sentence was included in the last paragraph of the Introduction: “The film characterization included Fourier-transform infrared (FTIR) spectroscopy, scanning electron microscopy, water vapor permeability, water contact angles, and shear force measurements.”

  1. Materials and Methods: The authors must separate in paragraphs this section. For example: 2.3 FTIR, 2.4 SEM, 2.5 WVP etc.

The 2.2 subsection (“Film characterization”) was divided into subsubsections (2.2.1 FTIR, 2.2.2 SEM etc).

  1. For WVP which is the equation used for the calculated values. PLs refer into the revised materials and results section.

The equation was included and explained (in 2.2.3).

  1. Make a seperate materials and method section for the statistical analysis used.

The statistical analyses was made as a separate subsection (2.3).

  1. Consider the addition of some antioxidant activity test of films (DPPH assay) as the tomato puree used gives some antioxidant activity to such films.

Unfortunately, we don’t have remaining film samples for a DPPH assay, and any film that we made now would be from different BC and tomato puree batches.

  1. Results and discussion: This section must also separate in section and re-organized to be more readable for readers. So, start first with FTIR and SEM characterization result section and continue with the other experimental results.

The Results and discussion section was re-organized (by starting with FTIR and SEM) and divided into subsections.

Reviewer 2 Report

The film prepared in this work (tomato puree/polysaccharides/palm olein) exhibited low water vapor permeability and could reduce the moisture migration from the guacamole to the nachos, thus helping to maintain the typical nacho crispiness. However, major revision are still needed. Some suggestions for modification are put forward as follows.

1.     “edible films” is suggested to be added as a keyword.

2. Figure 4 should be renumbered, for example named A1 and A2, it is inappropriate to use the same number to represent two different figures.

3. It would be better to offer some contact angle data of the films with different components to show the effect of components on hydrophilicity.

4. Water solubility or absorptivity is an important parameter of edible film. It would be better to offer some data.

5. How about the antibacterial property of those films?

6. The author stated in the Conclusion section that “Other applications should be tested”, which is obviously inappropriate and can be changed to “the film prepared in this work can also be used as an interlayer between pizza crust and toppings”, thus highlighting its practical application potential.

7. Bacterial cellulose is widely applied in foods, packing, biomedical applications, etc. More references are suggested to be cited for broad readers. For example, “Plant extract-loaded bacterial cellulose composite membrane for potential biomedical applications; Cellulose nanocomposites: Fabrication and biomedical applications; Electrospun Functional Materials toward Food Packaging Applications: A Review” are suggested to be cited.

8. Please check the “References”, some references need to be revised, for example pages are missing for Ref. 12/15/27.

Author Response

Response – Reviewer 2:

The film prepared in this work (tomato puree/polysaccharides/palm olein) exhibited low water vapor permeability and could reduce the moisture migration from the guacamole to the nachos, thus helping to maintain the typical nacho crispiness. However, major revision are still needed. Some suggestions for modification are put forward as follows.

We thank the reviewer for the comments, and hope this revised version is satisfactory.

  1. “edible films” is suggested to be added as a keyword.

It was added as a keyword.

  1. Figure 4 should be renumbered, for example named A1 and A2, it is inappropriate to use the same number to represent two different figures.

Figure 3 (former Figure 4) has been renumbered as suggested by the reviewer. Thanks!

  1. It would be better to offer some contact angle data of the films with different components to show the effect of components on hydrophilicity.

Actually, Table 1 presents contact angle values of films with all component proportions, so it’s possible to notice that the film with the highest palm olein content (film #3) presented a higher WCA when compared to the films with the highest tomato puree content (film #1) and the highest BC+CMC content (film #2), which made us state that “Palm olein was the component that most contributed to lower the WVP and to increase the contact angles (Table 1, Figure 4), which is ascribed to the hydrophobicity of palm olein.”

  1. Water solubility or absorptivity is an important parameter of edible film. It would be better to offer some data.

Actually, we tried to measure water solubility, but all films were completely disintegrated after 24 h of immersion, and that’s why we had not presented the results in the first version. However, we’ve described that in the revised version.

  1. How about the antibacterial property of those films?

We did not measure antibacterial properties, since there were no components with particularly promising antibacterial activity.

  1. The author stated in the Conclusion section that “Other applications should be tested”, which is obviously inappropriate and can be changed to “the film prepared in this work can also be used as an interlayer between pizza crust and toppings”, thus highlighting its practical application potential.

We changed the sentence to: “The film prepared in this study may also be used as a moisture barrier layer in other multicomponent foods, such as in pizzas (between the crust and the toppings).”

  1. Bacterial cellulose is widely applied in foods, packing, biomedical applications, etc. More references are suggested to be cited for broad readers. For example, “Plant extract-loaded bacterial cellulose composite membrane for potential biomedical applications; Cellulose nanocomposites: Fabrication and biomedical applications; Electrospun Functional Materials toward Food Packaging Applications: A Review” are suggested to be cited.

The first two suggested references have been added. On the other hand, the paper “Electrospun Functional Materials toward Food Packaging Applications: A Review” involves a variety of polymers for electrospinning, and bacterial cellulose is not one of them.

  1. Please check the “References”, some references need to be revised, for example pages are missing for Ref. 12/15/27.

Refs. 14 (former 12) and 17 (former 15) were fixed. Ref. 27 was also fixed, although it doesn’t have page numbers (but only a paper number – 528), since some journals do not present page numbers anymore.

Round 2

Reviewer 1 Report

The most requested revisions done. I understand the difficult to prepare films again! Accept in this form.

Best wishes!

Reviewer 2 Report

The manuscript is well revised according to the comments and could be accepted now.